# Dynamic Barrier Coverage in a Wireless Sensor Network for Smart Grids

**DOI:** 10.3390/s19010041

**Published:** 2018-12-22

**Authors:** Fei Fan, Qiaoling Ji, Gongping Wu, Man Wang, Xuhui Ye, Quanjie Mei

**Affiliations:** 1School of Power and Mechanical Engineering, Wuhan University, Wuhan 430072, China; gpwu@whu.edu.cn (G.W.); manwang@whu.edu.cn (M.W.); xhye@whu.edu.cn (X.Y.); meiquanjie@whu.edu.cn (Q.M.); 2School of Mechanical Engineering and Automation, Wuhan Textile University, Wuhan 430073, China

**Keywords:** barrier coverage, wireless sensor network, smart grid, inspection robot, delay-tolerant sensor network, real-time network, ordinal potential game, distributed algorithm

## Abstract

The development of engineering technology such as inspection robots (IR) for transmission lines and wireless sensor networks (WSN) are widely used in the field of smart grid monitoring. However, how to integrate inspection robots into wireless sensor networks is still a great challenge to form an efficient dynamic monitoring network for transmission lines. To address this problem, a dynamic barrier coverage (DBC) method combining inspection robot and wireless sensor network (WSN) is proposed to realize a low-cost, energy-saving and dynamic smart grid-oriented sensing system based on mobile wireless sensor network. To establish an effective smart grid monitoring system, this research focuses on the design of an effective and safe dynamic network coverage and network nodes deployment method. Multiple simulation scenarios are implemented to explore the variation of network performance with different parameters. In addition, the dynamic barrier coverage method for the actual scene of smart grid monitoring considers the balance between network performance and financial costs.

## 1. Introduction

In the past decades, the world witnessed the evolution of the smart grid. With the development of engineering technology, inspection robots (IR) and wireless sensor networks (WSNs) are more widely used in power grid monitoring system [1]. In our previous research [2], a robot delay-tolerant sensor network (RDTSN) has been proposed to solve the problem of remote monitoring of transmission lines. 

This novel mobile WSN combines the characteristics of IR, WSNs and distribution of transmission line to realize remote patrolling and monitoring of smart grid. In addition, RDTSN significantly improves the interrupt/delay tolerance and data fault tolerance of distributed WSN monitoring system under the premise of non-real-time data transmission. However, the real-time is the basic element of monitoring system, improving the real-time of RDTSN is conducive to improving network performance and transmission line monitoring quality. It is very important to effectively improve the coverage area of WSN, in order to reduce network delay [3].

Coverage is one of the most important issues in wireless sensor networks [4]. It determines the detection area of sensor nodes and the monitoring degree of the target. The application scenarios of each wireless sensor network have different coverage requirements. The problem is usually divided into three research directions, namely, target coverage, area coverage and barrier coverage (linear coverage) [5]. 

Target coverage is usually applied to a limited number of discrete target surveillance scenarios. It is a common layout method of nodes in distributed WSN monitoring system. In the area coverage, the dispersed sensor nodes are used to monitor the whole area of the network. In addition, partial coverage is another coverage known as barrier coverage. Target coverage is limited by only a small number of targets, and the number of sensor nodes deployed is very small. Area coverage requires a large number of nodes to meet the full coverage of the area. For the same coverage area, the number of nodes needed to form a barrier coverage is between the number of nodes required for target coverage and area coverage. For overhead high voltage transmission lines, the monitoring targets are distributed throughout the entire corridor. Compared with target coverage and area coverage, barrier coverage can better adapt to the linear distribution characteristics of transmission lines.

In this study, we address a new coverage problem named dynamic barrier coverage (DBC) problem as shown in Figure 1. DBC is very suitable for application in environmental monitoring of smart grid. Environmental monitoring of transmission lines with DBC can be divided into two categories: monitoring targets along the line and monitoring of environmental variables on both sides of the line (such as bird invasion). In DBC, a small number of mobile robots are moving through the linear barrier. Linear deployment of fixed sensor nodes is different from the continuity of conventional barrier coverage, which can achieve a delay-tolerant sensor network to address communication requirements due to discontinuous deployment. Deploying more sensors means higher hardware costs. Therefore, DBC based on application of transmission line selection can effectively reduce the deployment cost of wireless sensor networks. 

Compared with traditional target coverage, dynamic barrier coverage can detect anomalies earlier, thus allowing more response time. It requires fewer sensors than traditional area coverage, which in turn reduces deployment costs. However, traditional barrier coverage cannot be used in these applications because it builds a continuous barrier extending from left to right or from the top to bottom of the area, and the barrier constructed does not provide overall targets coverage. In summary, DBC is a more flexible, economical and reliable deployment and coverage scheme for sensor networks than traditional coverage methods. In addition to smart grid coverage, DBC is also applicable to other linear or circular environmental variables monitoring, such as river, industrial area pollution monitoring and mountain, forest meteorological detecting, etc.

In this paper, we propose a discrete heuristic method (dynamic coverage game algorithm, DCGA) based on game theory to solve this problem. Given the length of the linear monitoring area *l* or the number of targets *n_t_*, set the appropriate communication range and number of fixed sensors to *Rc* and $\hat{v}$*_i_*, respectively, of which $\frac{l}{Rc}$ ≥ $\hat{v}$*_i_*, *n_t_* ≥ $\hat{v}$*_i_*. For example, in the coverage of transmission line monitoring network, fixed sensor nodes can be deployed on various power towers and mobile inspection robots can walk on the transmission line. 

Firstly, a coverage graph (CG) is constructed based on the barrier coverage structure. Mobile and fixed sensor nodes and their communication structures and coverage can be represented in the CG. Then, the dynamic barrier coverage problem is modeled according to the information in Coverage Graph, and the relationship between network connectivity and coverage is summarized. The objective function and constraint conditions of DBC are proposed. Finally, a potential game is designed for DBC, the utility function and potential function are established according to the communication delay and energy consumption of each node. In the game, each node will execute deployment actions according to its own benefits and the information of the surrounding nodes, and eventually eliminate unnecessary sensors to achieve the Nash equilibrium (NE) of the node set. 

The main contributions of this paper are summarized as follows:A method of monitoring network coverage for transmission line or other linear applications is proposed.Modeling the deployed nodes with a network coverage graph.Mathematical modeling of dynamic barrier coverage and definition of constraints to meet network requirements are given.A distributed algorithm based on potential game is designed for the problem.

The remainder of this paper is organized as follows. Section 2 reviews about related literature and work. Section 3 introduces the system model and the problem statement. In Section 4, first, the proposed approach is modeled and formulated based on game theory; then, a learning algorithm is introduced to converge to NE. Simulation work and numerical results and discusses are presented in Section 5, and we conclude the paper in Section 6.

## 2. Related Work

This section briefly reviews the research work of barrier coverage in wireless sensor networks. The concept of barrier coverage originates from the requirement of multi-robot long-distance communication [3], and is gradually extended to the research and application of wireless sensor networks [4,5,6]. We summarize the main research contents and methods of barrier coverage, which not only provide preliminary knowledge in this field, but also point out the direction for further research on dynamic-barrier coverage.

**Sensor mobility:** The recent deployment of mobile sensors in WSN has aroused great interest. Building WSN in harsh environments using mobile sensors such as robots and unmanned aerial vehicles may be valuable. The barrier coverage considering sensor mobility [7,8,9,10,11,12] aims to effectively improve barrier coverage under the constraints of existing mobile sensors and their range of movement. Furthermore, this paper deeply analyzes the two roles of mobile sensor in barrier coverage, one is the influence of mobile sensor on sensor range and model, the other is the influence of mobile sensor on network connectivity and network delay. These studies are closely related to the solution to the DBC problem.

**Sensing model:** Boolean disk sensing model [13] is widely used in the study of barrier coverage problem [14,15,16,17,18,19], which is represented by omnidirectional circle. In this model, the probability of 0 or 1 indicates that the target is within or outside the sensor node's sensing range. There are different sensing models with different representations according to practical applications, such as the application of 3D sensing models in camera sensor networks [20]. In addition, some work [20,21,22] has proposed sensing models based on probability representation considering the influence of sensing distance, angle and noise. Considering the practical application, we use a mobility-based disk sensing model in our research.

**Deployment methods:** There are two main barrier coverage methods for deploying fixed sensor nodes: random deployment [23] and deterministic deployment [24]. The former involves scheduling randomly distributed sensor nodes in WSN to establish sensor barriers, while the latter aims to globally optimize the location of sensors to minimize the total number of sensors while ensuring the performance of barrier coverage [25]. For deterministic deployment, geometric-based, linear and curve deployment is proposed in [23,24,25,26]. We analyze the situation of random deployment and deterministic deployment, and propose a global optimal deployment method.

**Coverage strength:** Coverage strength is expressed by the probability of sensing target, the continuity of the barrier and the number of barriers. Common WSN barrier coverage strength can be classified as follows: K-barriers and single barrier [27], strong barrier coverage and weak barrier coverage [26], worst and best [28] coverage, etc. Based on the principle of sparse deployment of sensor nodes for transmission line monitoring, the single barrier based on network connectivity probability is studied in this paper.

**Energy consumption:** Restricting and reducing the energy consumption of nodes is one of the important goals of WSN. A great deal of researches on barrier coverage [29] have proposed many methods to balance network performance while reducing network consumption and prolonging network lifetime. The main method is to balance network energy consumption and prolong network lifetime by setting up redundant and wake-up machines [30]. Therefore, balancing network load and node energy consumption is an important research direction of the text, in order to achieve more effective barrier coverage in realistic scenarios.

**Coverage algorithm:** The solution of barrier coverage problem can be categorized as centralized and distributed algorithms. In the literature [31], WSN is considered as a whole, only the performance of the whole network is considered when analyzing and solving the problem of barrier coverage. However, such centralized algorithms often ignore the performance of individual nodes, while distributed algorithms focus on the performance analysis of each node, and change the performance of the entire network by adjusting the coverage of individual nodes. In this paper, a distributed algorithm based on game theory is proposed to achieve global optimization of network performance under the premise of guaranteeing the best benefits of each node.

In practical applications, barrier coverage is widely used in the defense military field, especially the border intrusion monitoring coverage problem [29]. In this paper, the application of barrier coverage extends to other important linear monitoring areas such as smart grids, we further extend the practical utility of barrier coverage. In the application of barrier coverage, network performance such as network coverage strength, energy consumption and real-time performance are the focus of everyone's attention. Summarizing and comparing the solutions of other coverage problems also have extremely reference value.

Target coverage is applied to the WSN coverage problem of discrete targets. The concept of target coverage is proposed for the first time by Cardei et al. [31], and it is proved that the problem is NP-complete. Since then, a variety of centralized and discrete algorithms have been used to solve the target coverage problem. Most of these methods are based on greedy algorithm [31], linear algorithm [32] and game theory [33]. 

Area coverage is the most common form of network coverage, and large number of literatures [34,35,36] have conducted in-depth researches on this issue. The main solution is based on game theory [34] and greedy algorithm [36]. With the introduction of barrier coverage, models covering problems tend to be more diverse and more complex. More abundant heuristic algorithms, such as learning automaton [37], imperialist competitive algorithm [26] and integer linear programming [35], are used to solve the coverage problem. 

To sum up, few previous studies have proposed a game theory framework to balance network energy consumption and coverage strength based on deployment and scheduling by agent autonomous decision. In addition, in most of the above studies, sensor deployment is fixed and predefined, and the increase in the number of inputs will increase the processing time and complexity of the algorithm. The distributed game theory method is proposed in this paper, using appropriate secondary deployment and dynamic scheduling methods to achieve efficient linear coverage for WSN. It also saves sensor energy and prolongs network lifetime. Therefore, the problem of coverage of the transmission line monitoring network based on barrier coverage is solved in a pioneering manner.

## 3. Models and Definitions

In this section, we formally define the DBC problem. Furthermore, the model and definition for analyzing DBC problems are proposed. It is assumed that the transmission line can be described as a belt shape monitoring area with a length of L and a width of W. In this area, n IRs are used to monitor transmission lines dynamically, while N fixed nodes are randomly deployed along the lines to establish communication links and monitor key areas. Obviously, due to the existence of dynamic nodes, WSN can achieve dynamic coverage of the monitoring area after a long enough monitoring time. This chapter focuses on how to improve the coverage of communication of fixed nodes to effectively reduce communication delay and improve the quality of smart grid monitoring. 

We define a virtual node set *V* = {*v*_1_, *v*_2_, …, *v_n_*}, and the nodes could satisfy K-barriers coverage [25] in the belt area. Optimal deployment method is adopted to reduce node redundancy and achieve the maximum coverage of the minimum number of nodes, so as to improve the practical application value. Thus, we introduce some definitions here to better describe the problem.

### 3.1. Related Problem Formulation

**Definition** **1. Dynamic sensing model.**
*In the dynamic sensing model, the sensing area of the mobile node is dynamic, and the coverage area is directly related to the movement of the node, i.e., satisfiable Equation (1),*
(1)$$S_{c}(t_{o},t_{c},t_{s})=2R_{s}+v_{IR}*(t_{s}+t_{c}-t_{o})$$

*A S_c_ is a dynamic sensing coverage area. The onset time, sensing duration and communication delay are t_o_, t_s_ and t_c_ respectively. In addition, v_IR_ is the average rate of IR movement, R_s_ is robot sensing range. With the mobile node IR moving on transmission lines, the whole smart grid is in its dynamic sensing range.*


**Definition 2.** **Communication coverage model.**
*In the communication coverage model, sensors adopt omnidirectional disk communication model, and the communication coverage of each node is expressed by Boolean variables. The specific model is shown in Equation (2),*
(2)$$C_{i}(v_{i},p)=\{\begin{array}{c} 1\\ 0 \end{array}\begin{array}{c} ,\\ , \end{array}\begin{array}{c} if\\ if \end{array}\begin{array}{c} d(v_{i},p)\leq R_{c}\\ d(v_{i},p)>R_{c} \end{array}$$

*In Equation (2), if the event or target is within the communication coverage R_c_ of node v_i_, the probability that node v_i_ can be connected at that location is 1, otherwise, the value is 0. In addition,*
$d\left(v_{i},p\right)=\sqrt{{\left(x_{i}-x_{p}\right)}^{2}-{\left(y_{i}-y_{p}\right)}^{2}}$
*is Euclidean distance between the ith node V_i_ (x_i_, y_i_) and the location of occurring event or target, which is P (x_p_, y_p_).*


**Definition 3.** **Coverage Graph (CG).**
*The Coverage Graph CG = (V, E) of mobile wireless sensor networks is as follows: vertex set V corresponds to all sensor nodes. The mobile node in the network is the source node, then, there are several target nodes in the left and right edges of the network and in the middle of the network links (For example, the target node in the middle and both ends of the network link of Figure 2). There exists an edge E between two nodes, if these nodes are located between the communication ranges Rc of each other.*


The accurate establishment and research of dynamic barrier can be guaranteed by CG. In detail, in a dynamic barrier, CG is used to check connectivity among adjacent nodes and from source nodes to destination nodes. Figure 2 shows a typical coverage graph of DBC. Because of the presence of robots in the graph, a dynamic network link is constructed without the intersection of all nodes at different time. The links without intersection between nodes can be represented by virtual network links. The virtual network link indicates that the robot in the network carries and transmits information on the move to ensure that the message in the network completes delivery to the destination with a certain delay. The path of message transmission in the network is called dynamic barrier path, and its detailed definition can be found in Section 3.2.

### 3.2. Formal Definition for Dynamic Barrier Coverage Problem

**Definition 4.** **Dynamic Barrier Path (DBP).**
*A DBP is a continuous or discontinuous horizontal or vertical path composed of a series of sensor nodes. The sensing area of the dynamic node should cover the whole path, and the communication area between the fixed node and the adjacent sensor nodes should overlap locally. A DBP can be defined as follows,*
(3)$$DBP(v_{x})=\overset{k}{\underset{i=1}{\cup }}v_{i}$$
*where k is the number of nodes in each DBP, v_i_∈ V is fixed node within DBP.*


In the dynamic monitoring network of smart grid, dynamic monitoring should be done by DBP, which can transmit data to external network through multi-hop communication. The main goal of DBC problem is to find the best DBP by distributed method based on game theory. In a network with size of *L***W*, a DBP consists of N randomly deployed fixed nodes and n mobile nodes. A dynamic monitoring link can be expressed as a subset of nodes *v_i_*
∈
*V*, so as to minimize the number of nodes within DBP, while ensuring network connectivity and coverage requirements.

Assuming that the probability that each *v_i_* of DBP has coverage overlap at least with another is *P*(*v_i_*
∩
*v_i_*_+1_
≠ 0), which represents the connectivity between two neighbor nodes. The connectivity of each node is different in a DBP, so the intersection of their coverage areas should satisfy the following constraints,
(4)$$P(v_{i}\cap v_{i+1}\neq 0)\in \left(0,1\right]$$

The connectivity of each node *v_i_*
∈
*V* in the dynamic barrier path (DBP) is different and the network link of the mobile node is dynamic. In the CG, as shown in Figure 3, a DBP contains multiple source nodes (robots) and target nodes, the number of which is *n*, *m*, respectively. Where the *j*th mobile node is *v_Sj_*, and the *k*th destination node is *v_Tk_*. The sensor nodes deployed along the transmission line can form a plurality of pending DBPs as shown in Figure 3. We assume that *v_Sj_* can connect to any fixed node in a DBP at different time. The dynamic node *v_Sj_* and the target node *v_Tk_* should satisfy the following constraints,
(5)$$P(v_{Sj})=\frac{\sum_{t_{o}}^{t_{s}}P(v_{Sj}\cap v_{i}\neq 0)}{t_{s}-t_{o}}\in \left(0,1\right]$$
(6)$$P(v_{Tk})=\max P(v_{i}\cap v_{Tk}\neq 0\cap v_{-i}\cap v_{Tk}\neq 0)\in \left(0,1\right]$$
(7)$$P(v_{Sn},v_{Tm})=\prod_{j=1}^{n}P(v_{Sj})*\prod_{k=1}^{m}P(v_{Tk})\in \left(0,1\right]$$

Equations (5) and (6) respectively show the connectivity probabilities of the source node *v_Sj_* and the target node *v_Tk_* for the static nodes, respectively. Equation (5) calculates the dynamic connectivity probability from the source node to the static node, where the value of the probability is represented by the average of the time dimensions.

As shown in Equation (6), the connectivity probability of the target node to the static node can be calculated by the connectivity between it and its two neighbor nodes. Where *i* represents the *i*-th static node *v_i_*, and −*i* represents the removal of *v_i_*, i.e., the remaining N-1 static nodes. The dynamic connectivity probability from the source node to the target node is calculated in Equation (7). And the connectivity probability of the entire DBP can be calculated by the connectivity between two adjacent nodes. It not only expresses the connectivity of DBP, but also reflects the communication delay of dynamic sensor networks. The connectivity probability of the DBP can be expressed by Equation (8),
(8)$$P(DBP^{k})=P(v_{Sn},v_{Tm})*\prod_{i=1}^{k-1}P(v_{i}\cap v_{i+1}\neq 0)\in \left(0,1\right]$$

**Definition 5.** **Dynamic Barrier coverage (DBC).**
*Dynamic barrier coverage problem is to find a solution to minimize the number of nodes in the optimized DBP. Meanwhile, the solution can also meet connectivity requirements of the DBP under different scenarios.*

*The objective function of DBC is as follows,*
(9)$$DBP_{\min}^{k}=\min{\hat{v}}_{i}$$
*where*
$\hat{v}$
*_i_ represents the number of fixed nodes in DBP. The objective function can reduce the complexity of the communication link calculation and the network delay of the dynamic monitoring information, and improve the real-time performance of the monitoring data.*


Boolean variable *κ_vi_* is used to indicate whether *v_i_*
∈ V is used to participate in a DBP formation, the specific expression is as follows,
(10)$$\kappa_{v_{i}}=\{\begin{array}{c} 1\\ 0 \end{array}\begin{array}{c} ,\\ , \end{array}\begin{array}{c} if \end{array}\begin{array}{c} v_{i}\in DBP^{k}\\ otherwise \end{array}$$

**Theorem** **1.**
*Consider the network area of size L * W, where L and W respectively represent the length and width of the network area. The relationship between the number of fixed nodes required for dynamic barrier coverage and DBP connectivity can be seen from the following Equation (11),*
(11)$$\{\begin{array}{c} \sum_{\forall v_{i}\in V}\kappa_{v_{i}}<\left\lceil \frac{L}{2R_{c}}\right\rceil \\ \sum_{\forall v_{i}\in V}\kappa_{v_{i}}\geq \left\lceil \frac{L}{2R_{c}}\right\rceil \end{array}\begin{array}{c} ,\\ , \end{array}\begin{array}{c} if\\ if \end{array}\begin{array}{c} P(DBP^{k})<1\\ P(DBP^{k})=1 \end{array}$$

*It can be inferred from the theorem 1 that in linear deployed WSNs, if there is no overlap between deployed nodes, the connectivity probability of DBP is less than 1, that is, DBP is constructed as an interrupt/delay-tolerant network [2]. Else if there is overlap between deployed nodes, DBP is a real-time network. In practical applications, we can adjust the network delay performance by changing the deployment density and location among nodes according to different scenario requirements. Then, the following theorem is presented to analyze the computational complexity.*


**Theorem** **2.**
*The problem of finding the optimized DBP is NP-hard.*


**Proof** **of Theorem 2.**By definition, DBP is a dynamic WSN composed of different sensors. From CG, we need to compute the set of vertices that satisfy the minimum weights of the constraints (application requirements). Therefore, we turn the problem into the minimum vertex cover problem in graph theory to solve it.Suppose CG = (*V*, *E*) is a weighted graph, where *E* = (*v_i_*, *v_i_*_+1_) e is the weight in the graph. The weight of each vertex is determined by the sensor location, energy consumption and the current constraints (network connectivity), that is, each vertex *v_i_*
∈
*V* has a weight *w*(*v_i_*). For example, the weight of each vertex is 1.Now, the unweighted graph CG′ can be regarded as a special case of the weighted graph CG. Then, the problem can be expressed as whether there is a set covering the whole CG′, and the number of vertices is the minimum. Therefore, the problem is just a minimum vertex cover problem. Since the minimum vertex cover problem is NP-complete, then the DBC problem is NP-hard.

## 4. The Proposed Method

In this section, a novel approach based on game theory is proposed to address the problem of barrier coverage for dynamic monitoring systems. This method proved to be a potential game and a heuristic solution to the DBC problem. It can converge into Nash equilibrium using a new distributed learning algorithm.

### 4.1. Rational Game Theory

In the game of DBP formation, each sensor, as a participant, selfishly wants to maximize its own benefits based on local information gathered from the network and actions taken by other participants. The effective benefits of each participant depend not only on its own behavior, but also on the actions taken by other participants.

When each participant chooses its own optimal action strategy, no one will actively deviate from its current strategy choice when other participants’ strategy remains unchanged. At this time, the strategy combination of each player is called Nash equilibrium. In addition, by designing an appropriate utility function, each sensor gets the best benefit, and the DBC problem can also obtain the global optimal solution.

**Definition 6.** **Nash equilibrium [38].**
*Suppose a strategy game Γ (V, S, u) has a combinatorial strategy*
$S_{i}^{*}$
*(*
si∗
*,*
$s_{-i}^{*}$
*)*
∈
*S for*
∀
*v_i_*
∈
*V, i*
∈
*N and if there is a s_i_*
∈
*S for*
∀
*v_i_*
∈
*V satisfies Equation (12), the combinatorial strategy*
$S_{i}^{*}$
*(*
si∗
*,*
$s_{-i}^{*}$
*) is a Nash equilibrium of the game Γ,*
(12)$$u_{i}(S_{i}^{*})\geq u_{i}(s_{i},s_{-i}^{*})$$
*where i represents participant i, and −i represents the remaining N-1 participants of i.*


A game may have more than one equilibrium, or it may not exist at all. Some types of games have at least one Nash equilibrium. In order to guarantee the existence of Nash equilibrium, a special kind of strategic game-potential game is proposed in [39], and it is proved that there exists at least one Nash equilibrium.

**Definition 7.** **Ordinal Potential Game (OPG) and Ordinal Potential Function (OPF) [39].**
*A strategy game Γ (V, S, u) is an OPG if there is a function O (S) that satisfies Equation (13). And the function O (S) is an OPF,*
(13)$$\begin{array}{l} u_{i}(s_{i1},s_{-i}^{*})>u_{i}(s_{i2},s_{-i}^{*})\\ \Leftrightarrow O_{i}(s_{i1},s_{-i}^{*})>O_{i}(s_{i2},s_{-i}^{*}) \end{array}$$


**Theorem 3** **[39].**
*If the strategy game Γ (V, S, u) is an OPG and O (S) is its OPF, the combinatorial strategy*
$S_{i}^{*}$
*(*
si∗
*,*
$s_{-i}^{*}$
*) that maximizes O (S) is a Nash equilibrium of the game Γ.*


Therefore, if the ordinal potential function of the strategy game is determined, the maximization of the ordinal potential function can be obtained by the combination strategy $S_{i}^{*}$ to obtain the Nash equilibrium of the strategy game.

### 4.2. Dynamic Barrier Coverage Game Model (DBCGM)

The dynamic barrier coverage game model (DBCGM) based on ordinal potential game is established in this section. In this model, each sensor chooses actions based on its own interests to form DBP. The action (strategy) of each player (sensor) is defined by a vector *s_i_* as follows: *s_i_* = *κ_vi_*
∈
*S_i_*. A DBP can be regarded as a set of actions *S_i_* = (*s*_1_, *s*_2_, …, *s_n_*) ∈
*S* that all players participate in and act on their own interests. It can be expressed as *S_i_* = (*s_i_*, *s_−i_*), where *s_−i_* = (*s*_1_, . . ., *s_i_*_−1_, *s_i_*_+1_, . . ., *s_n_*) refers to all actions taken by all nodes but *v_i_*.

Assume that when node *v_i_* takes (deployment/non-deployment) action, the number of deployment actions performed by itself and other nodes within its communication range is *n_i_* (*S_i_*) and *n_j_* (*S_i_*), respectively. Now, the definition of *n_i_* (*S_i_*) and *n_j_* (*S_i_*) for each fixed sensor node in a DBP is as follows,
(14)$$n_{i}(S_{i})=n_{i}(s_{i},s_{-i})=\kappa_{v_{i}},\;i\in N,\;v_{i}\in V$$
(15)$$n_{j}(S_{i})=n_{j}(s_{i},s_{-i})=\sum_{i\neq j,\forall v_{j}\in V}\kappa_{v_{j}},\;i\neq j,\;v_{i},v_{j}\in V,\;C_{i}(v_{i},v_{j})=1$$

**Definition 8.** **Coverage Connectivity Factor (CCF).**
*The CCF for each DBP is ρ*
∈
*(0,1], which can be preset by different application scenarios. ρ represents the expected goal of connectivity probability for each DBP.*


The connectivity probability of DBP under different combination strategies satisfies Equation (16). If *P* (*s_i_*, *s_−i_*) = 1, the network is connected, that is, node i can communicate with all other nodes through a two-way link, else if *P* (*s_i_*, *s_−i_*) ∈ (0,1] network shows dynamic intermittent connectivity. Obviously, *P* (*s_i_*, *s_−i_*) is a monotone nondecreasing function, that is, for any node *v_i_*, as it is deployed or removed (*s_i_*_1_ = 1 > *s_i_*_2_ = 0) in DBP, there is *P* (*s_i_*_1_, *s_−i_*) ≥ *P* (*s_i_*_2_, *s_−i_*),
(16)$$P(s_{i},s_{-i})=P(DBP^{{\hat{v}}_{i}}),\;\forall v_{i}\in V$$

When nodes transmit messages, the weight average of residual energy for sensors in DBP is given by Equation (17). Where *E_d_* (*v_i_*) is the residual energy of sensor *v_i_*
∈
*V*, and *E_o_* (*v_i_*) is the initial energy of sensor *v_i_*
∈*V*,
(17)$$E(v_{i,-i})=\frac{1}{k}\sum_{i}^{k}\frac{E_{d}(v_{i})}{E_{o}(v_{i})}\;\forall v_{i}\in V,\;i,k\in N$$

The most important step of solving the DBC problem is to determine the utility function of each node *v_i_*
∈
*V*, *i*
∈
*N*. The utility function represents the trade-off between the benefits and the cost of each node deploying and connecting to the network. For each DBP, the utility function for the *i*th node is defined as,
(18)$$u_{i}=P(s_{i},s_{-i})*\left[\frac{\alpha *n_{i}(S_{i})}{n_{i}(S_{i})+\frac{1}{P(S_{i})}*{\left(n_{j}(S_{i})-2\rho \right)}^{2}}+(1-\alpha )*E(v_{i,-i})\right]$$
where *α*
∈ (0,1] represents the weighting parameter pertinent to the cost and profit. As mentioned above *P* (*s_i_*, *s_−i_*) ∈ (0,1] is connectivity probability of DBP, *P* (*s_i_*, *s_−i_*) is positively correlated with the utility function. The greater the value of *P* (*s_i_*, *s_−i_*), the better the connectivity probability of DBP, and the higher the revenue of static nodes. In the utility function, the product term of *α* represents the gain of sensor *v_i_*
∈
*V* under different combination strategy *S_i_* = (*s_i_*, *s_−i_*). Under different application premises, by adjusting the size of CCF *ρ*, the sensor income value is changed, so that the performance of each node is closely related to the application target. 

The remaining part of Equation (18) describes the energy consumption of the sensor under the combined strategy *S_i_*. *E* (*v_i, −i_*) in the utility function indicates that a node tends to take the node with more residual energy as its neighbor to increase the value of the utility function in order to improve the average residual energy of its neighbor nodes. Energy consumption equalization is very important for optimizing the network topology of maximizing network lifetime. If the energy consumption of the nodes in the network is very uneven, some nodes will run out of energy quickly, which will lead to early termination of the entire network lifetime. The selection strategy based on this utility function helps to balance energy consumption between nodes. In addition, the setting of parameter *α* can further adjust the proportion of energy consumption to node income. Usually, we use *α* = 0.5, that is, the way of balancing income and cost to evaluate the node combination strategy.

**Theorem** **3.**
*The game model of DBC problem is an ordinal potential game. Its ordinal potential function is defined as,*
(19)$$O_{i}=\sum_{i\in N}\left\{P(s_{i},s_{-i})*\left[\frac{\alpha *n_{i}(S_{i})}{n_{i}(S_{i})+\frac{1}{P(S_{i})}*{\left(n_{j}(S_{i})-2\rho \right)}^{2}}+(1-\alpha )*E(v_{i,-i})\right]\right\}$$


**Proof** **of Theorem 3.**According to the utility function defined in Equation (19), it is possible to assume that *S_i_*_1_ and *S_i_*_2_ are two different strategic actions (*s_i_*_1_ = 1 > *s_i_*_2_ = 0 or *s_i_*_1_ = 0 < *s_i_*_2_ = 1) of node *i*, and the difference between the returns of node i when it’s selecting *S_i_*_1_ and *S_i_*_2_ respectively is as follows,
(20)$$\begin{array}{l} \Delta u_{i}=P(s_{i1},s_{-i})*\left[\frac{\alpha *n_{i}(S_{i1})}{n_{i}(S_{i1})+\frac{1}{P(S_{i1})}*{\left(n_{j}(S_{i1})-2\rho \right)}^{2}}+(1-\alpha )*E(v_{i,-i})\right]\\ -P(s_{i2},s_{-i})*\left[\frac{\alpha *n_{i}(S_{i2})}{n_{i}(S_{i2})+\frac{1}{P(S_{i2})}*{\left(n_{j}(S_{i2})-2\rho \right)}^{2}}+(1-\alpha )*E(v_{i,-i})\right]\\ =(1-\alpha )*E(v_{i,-i})*\left[P(s_{i1},s_{-i})-P(s_{i2},s_{-i})\right]+\frac{\alpha *n_{i}(S_{i1})*P(s_{i1},s_{-i})}{n_{i}(S_{i1})+\frac{1}{P(S_{i1})}*{\left(n_{j}(S_{i1})-2\rho \right)}^{2}}\\ -\frac{\alpha *n_{i}(S_{i2})*P(s_{i2},s_{-i})}{n_{i}(S_{i2})+\frac{1}{P(S_{i2})}*{\left(n_{j}(S_{i2})-2\rho \right)}^{2}} \end{array}$$

Similarly, when node *v_i_*
∈
*V*, *i*
∈
*N* selects *S_i_*_1_ and *S_i_*_2_ separately, the difference between its ordinal potential functions *Oi* is as follows,
(21)$$\begin{array}{l} \Delta O_{i}=\sum_{i\in N}\left\{P(s_{i1},s_{-i})*\left[\frac{\alpha *n_{i}(S_{i1})}{n_{i}(S_{i1})+\frac{1}{P(S_{i1})}*{\left(n_{j}(S_{i1})-2\rho \right)}^{2}}+(1-\alpha )*E(v_{i,-i})\right]\right\}\\ -\sum_{i\in N}\left\{P(s_{i2},s_{-i})*\left[\frac{\alpha *n_{i}(S_{i2})}{n_{i}(S_{i2})+\frac{1}{P(S_{i2})}*{\left(n_{j}(S_{i2})-2\rho \right)}^{2}}+(1-\alpha )*E(v_{i,-i})\right]\right\}\\ =(1-\alpha )*E(v_{i,-i})*\left[P(s_{i1},s_{-i})-P(s_{i2},s_{-i})\right]+\frac{\alpha *n_{i}(S_{i1})*P(s_{i1},s_{-i})}{n_{i}(S_{i1})+\frac{1}{P(S_{i1})}*{\left(n_{j}(S_{i1})-2\rho \right)}^{2}}\\ -\frac{\alpha *n_{i}(S_{i2})*P(s_{i2},s_{-i})}{n_{i}(S_{i2})+\frac{1}{P(S_{i2})}*{\left(n_{j}(S_{i2})-2\rho \right)}^{2}}+(1-\alpha )*E(v_{k,-k})*\sum_{i\neq k,k\in N}\left[P(s_{k1},s_{-k})-P(s_{k2},s_{-k})\right]\\ +\sum_{i\neq k,k\in N}P(s_{k1},s_{-k})*\frac{\alpha *n_{k}(S_{k1})}{n_{k}(S_{k1})+\frac{1}{P(S_{k1})}*{\left(n_{j}(S_{k1})-2\rho \right)}^{2}}\\ -\sum_{i\neq k,k\in N}P(s_{k2},s_{-k})*\frac{\alpha *n_{k}(S_{k2})}{n_{k}(S_{k2})+\frac{1}{P(S_{k2})}*{\left(n_{j}(S_{k2})-2\rho \right)}^{2}} \end{array}$$

Therefore, the following Equation (22) can be obtained,
(22)$$\begin{array}{l} \Delta O_{i}=\Delta u_{i}+(1-\alpha )*E(v_{k,-k})*\sum_{i\neq k,k\in N}\left[P(s_{k1},s_{-k})-P(s_{k2},s_{-k})\right]\\ +\sum_{i\neq k,k\in N}P(s_{k1},s_{-k})*\frac{\alpha *n_{k}(S_{k1})}{n_{k}(S_{k1})+\frac{1}{P(S_{k1})}*{\left(n_{j}(S_{k1})-2\rho \right)}^{2}}\\ -\sum_{i\neq k,k\in N}P(s_{k2},s_{-k})*\frac{\alpha *n_{k}(S_{k2})}{n_{k}(S_{k2})+\frac{1}{P(S_{k2})}*{\left(n_{j}(S_{k2})-2\rho \right)}^{2}} \end{array}$$

Since *P* (*s_i_*, *s_−i_*) is a nondecreasing function, the following Equation (23) is obtained,
(23)$$\begin{array}{l} \Delta u_{i}\{\begin{array}{c} >0\\ <0 \end{array}\begin{array}{c} \begin{array}{cc} & if\begin{array}{cc} & s_{i1}=1>s_{i2}=0 \end{array} \end{array}\\ \begin{array}{cc} & if\begin{array}{cc} & s_{i1}=1>s_{i2}=0 \end{array} \end{array} \end{array}\\ \Leftrightarrow \Delta O_{i}\{\begin{array}{c} >0\\ <0 \end{array}\begin{array}{c} \begin{array}{cc} & if\begin{array}{cc} & s_{i1}=1>s_{i2}=0 \end{array} \end{array}\\ \begin{array}{cc} & if\begin{array}{cc} & s_{i1}=1>s_{i2}=0 \end{array} \end{array} \end{array} \end{array}$$

In summary, when the initial condition is the same (*s_i_*_1_ = 1 > *s_i_*_2_ = 0 or *s_i_*_1_ = 0 < *s_i_*_2_ = 1), *sgn* (△*u_i_*) = *sgn* (△*O_i_*), i.e., DBCGM, the game model of DBC problem is an ordinal potential game. Its ordinal potential function is *O_i_*.

**Inference** **1.**
*There must be Nash equilibrium in the Dynamic Barrier Coverage Game Model (DBCGM).*


**Proof** **of Inference 1.**According to Theorem 3, the function *O_i_* shown in Equation (19) is exactly the ordinal potential function of DBCGM. As can be seen from Theorem 2, the combined strategy of maximizing its ordinal potential function is the Nash equilibrium solution of the model. And any node *v_i_* can choose whether to participate in the formation of DBP or not, i.e., the optional strategy *s_i_* of player *v_i_* is limited, so there must be a combination of strategies *S_i_* = (*s_i_*, *s_−i_*) that maximizes *O_i_*. This combination of strategies is the solution of the DBCGM.

### 4.3. Distributed Learning Algorithm

On the one hand, the utility of each player (sensor) is based on the actions taken by itself and other players (especially their neighbors) and their energy performance. On the other hand, since the initial state of network connectivity is uncertain, we assume that each sensor only knows the state of other nodes in its communication range. Therefore, due to the above information constraints, the nodes are unable to calculate the global payoff of a DBP accurately, which associated with alternative actions of each sensor. In addition, on the entire DBP link, only the latest actions that can be contacted locally are available for utility value calculations. Therefore, developing a learning algorithm based on distributed payment is a good option for DBCGM.

In order to solve DBCGM in a distributed way, according to the detailed description of the execution steps of the distributed algorithm, the distributed coverage game algorithm (DCGA) which controls node deployment needs to satisfy the following assumptions:(1)All nodes in the network have unique IDs, which increase from one end of a DBP to the other according to the distribution of DBPs.(2)The connectivity of the whole DBP can be judged in a distributed way.(3)Each sensor can obtain the IDs of one hop neighbor node, the remaining energy and the number of other nodes within the communication range of it and other information.

For condition (1), the nodes can be sorted by the programming before deployment; for condition (2), the node can complete acquisition of the required information by broadcasting to the neighbor node; for condition (3), the connectivity of different DBPs can be calculated by using the relevant Equations (4)–(9) in Section 3 of this paper in combination with the distributed method.

The distributed coverage game algorithm (DCGA) includes the following three steps:

Step 1: Neighbor discovery 

Obtain information about itself and all neighbor nodes within the communication range of the node, including the ID of the node, remaining energy, initial energy, node location, number of neighbor nodes, etc.

Step 2: Game execution 

Each node selects an appropriate deployment action strategy and performs a game based on the state information of itself and the neighbor nodes, such as the initial energy and remaining energy of the node, the number of neighbor nodes and the gaming behavior of the neighbor nodes, etc.

Step 3: Network maintenance

During the execution of the monitoring task, the DBP dynamically adjusts the deployment plan and network structure of the fixed nodes in the network according to the running status of each node in the network. In this phase, DBP will re-execute the neighbor discovery process and the game execution process according to a periodic time, until stopping criteria is met.

The implementation process of each stage is described in detail in the following sections.

#### 4.3.1. Neighbor Discovery Phase

In DCGA, the state of the network is closely related to the running time of the network. The distributed learning algorithm proposed in this paper takes discrete time as triggering condition. At initial time t = 0 of the neighbor discovery process, each node *v_i_* initializes its own action, which satisfies Equation (24),
(24)$$S_{i}(t=0)=(s_{i}(t=0),\;s_{-i}(t=0))=(1,1)$$

The process involves the following steps:

Step 1: At t = 0, all nodes broadcast their IDs and other information to the outside.

Step 2: Each node adds the accepted broadcast information to the neighbor list, including the number of neighbors, initial energy, remaining energy, node location, etc.

Step 3: Each node will turn off the broadcast when t = 1. And each node generates its own set of pending strategies *S_i_* = (*s_i_*, *s_j_*_1_, *s_j_*_2_, …, *s_jk_*), where *i*, *j*1, *j*2, …, *jk*
∈
*N* and *k* is the number of one-hop neighbor nodes *j* of node *i*, through information exchange.

The pseudo code of neighbor discovery process is as follows.

**Algorithm 1.** DCGA for every node in the network in neighbor discovery phase1: node start (initialization)2: **if**
*t* = 0 **then**3: broadcast ID4: listen for neighbors’ IDs5: wait for messages ()6: **if** message receives in queue **then**7:  update number of neighbors8:  update location of neighbors9:  update initial energy of neighbors10:  update residual energy of neighbors11: **end if**12: **end if**13: **if**
*t* = 1 **then**14: Turn off the broadcast and listen15: generates its own set of pending strategies *S_i_* = (*s_i_*, *s_j_*_1_, *s_j_*_2_, …, *s_jk_*)16: **end if**

#### 4.3.2. Game Execution Phase

During the execution of the game, all nodes determine their deployment actions by randomly or alternately executing the game according to the node ID serial number. Each round has only one node to adjust the game strategy, and the other nodes remain unchanged. While game, the strategy and utility value update rules of sensors are as follows,
(25)$$t_{x}=\{\begin{array}{c} t\\ t-1 \end{array}\begin{array}{c} ,\\ , \end{array}\begin{array}{c} if \end{array}\begin{array}{c} u_{i}(s_{i}(t))\geq u_{i}(s_{i}(t-1))\\ otherwise \end{array}$$
where *t_x_* is the best moment for the agent *v_i_*
∈
*V* to get the best income at the last two steps. As described above, at time *t* = 0 and *t* = 1, all nodes are in an active and closed state, respectively. In the stage of performing the game (update), i.e., *t* ≥ *2*, each agent updates their status based on the following rules:(1)Sensor *v_i_* determines whether to enter the exploration or exploitation stage by the exploration rate *ε*(*t*) = $\frac{1}{\ln t}$.(2)With a probability of *P* = *ε*(*t*), the sensor *v_i_* enters the exploration stage and tests random actions. Then, there are two situations:
Case 1: *s_i_* (*t* − 1) = 0 → *s_i_* (*t*) = 1Case 2: *s_i_* (*t* − 1) = 1 → *s_i_* (*t*) = 0(3)With a probability of *P* = 1 − *ε*(*t*), the sensor *v_i_* enters the exploitation stage and sets its action to $s_{i}^{act}$ = *s_i_* (*t_x_*).(4)When $s_{i}^{act}$ is determined, the strategy action will determine whether sensor *v_i_* is performing deployment or being removed. If $s_{i}^{act}$ = 1, the sensor will be deployed in DBP. Otherwise, it will be removed from it.

The pseudo code of optional game execution phase is as follows.

**Algorithm 2.** DCGA for every node in the network in game execution phase.1: **if**
*t* > 2 **then**2: Setting and Calculating *ε*(*t*) = $\frac{1}{\ln t}$3: **if**
*P* = *ε*(*t*) **then**4:  **switch** (*s_i_* (*t* − 1))5:   **Case 1:** 0 → *s_i_* (*t*) = 16:   **break**7:   **Case 2:** 1 → *s_i_* (*t*) = 08:   **break**9: **end if**10: **if**
*P* = 1 − *ε*(*t*) **then**11:    Calculating *s_i_* (*t_x_*)12:    $s_{i}^{act}$ ← *s_i_* (*t_x_*)13: **end if**14: **return**
$s_{i}^{act}$15: **end if**

#### 4.3.3. Network Maintenance Phase

In this phase, each sensor *v_i_*
∈
*V* performs its strategic action $s_{i}^{act}$. And its status information including *E_d_* (*v_i_*), *E_o_* (*v_i_*) and $s_{i}^{act}$ are sent to its neighboring nodes (sensors whose distance from node *v_i_* is less than or equal to its communication radius). Now, the entire DBP realizes real-time or delay-tolerant communication network through interaction of dynamic sensors with other sensors. And the connectivity probability *P_i_* ($s_{i}^{act}$, $s_{-i}^{act}$) of the DBP can also be calculated. Each sensor *v_i_* then calculates its utility *u_i_* ($s_{i}^{act}$, $s_{-i}^{act}$) based on the data interacting from neighbors in the DBP.

In this phase, DBP will repeats the steps above according to a periodic time, until stopping criterion is met.

The stopping criterion for this algorithm (DCGA) contains two cases. One is that the connectivity of the DBP satisfies the preset coverage connectivity factor (the numerical error of both is less than 5%). The other is affected by the initial deployment factor of the node, which makes DBP connectivity impossible to meet the requirements of the application. Then, the algorithm will continue to run to 1000 iterations to prevent the absence of the best solution.

## 5. Performance Evaluation

This section uses an opportunistic network environment (ONE) [40] (ONE version 1.6.0 developed by the University of Alto, Finland, 2015) and MatTuGames (MATLAB Game Theory Toolbox) to achieve a co-simulation. Through simulation experiments, the performance of Distributed Coverage Game Algorithm (DCGA) for transmission line monitoring is obtained.

This simulation experiment establishes a robotic delay-tolerant sensor network (RDTSN) [2] for an overhead high voltage transmission line monitoring, which located in Jilin Province, China. Then a simulation test of dynamic barrier coverage is performed for the RDTSN. This is a 220-kV overhead high-voltage transmission line with a length of about 120 kilometers. It crosses the primordial forest and extends to the border areas of China and North Korea. This article uses the Open Street Map [41] to obtain local geographic and traffic information. In addition, the moving path (transmission line) of the inspection robot is depicted in the map. Firstly, according to the transmission line information and related settings, the distributed coverage game algorithm (DCGA) calculation result is obtained through MatTuGames, and the static node coordinates to be deployed are derived. 

For simulation scenarios, the coverage area of the network is set to 74,500 (m) × 143,400 (m). All nodes have wireless communication capabilities similar to IEEE 802.11n, and the transmission rate is 8 Mbps. There are three kinds of nodes in the network: source nodes (inspection robot), target nodes (Wireless Central Nodes, WCNs), static nodes (SNs). The default network transmission range of WCNs, SNs and robot is 3 km. The default buffer size of the node is 500 M. The default number of robots is 5, the default motion model is BPRW [2], and the motion rate is 3–5 m/s. At BPRW model, robot takes the time of *T_pause_* ∈ (*T_min_*, *T_max_*) to remain stationary or complete an obstacle-climbing process. The initial battery capacity of WCN and SN is 100 Ah, and the initial battery capacity of the robot is 57 Ah. The message generation of each node follows the Poisson process. The message arrival time of robot message is 30–120 s, and the message size is 2 M. The above parameters are based on actual robot and transmission line characteristics. Each simulation run time is 86,400 s, where the 24-h running results of actual network was simulated. 200 simulations were carried out for each scenario. The default simulation parameters are shown in Table 1.

### 5.1. Performance of DCGA with Different Parameters of ρ

In order to determine the influence of various parameters on network performance, this paper designs a simulation scenario 1 to adjust the relevant tunable parameters (coverage connectivity factor *ρ*) and try to find the optimal configuration. In simulation scenario 1, the default value of the coverage connectivity factor *ρ* is changed. The value of *ρ* parameter varies from 0 to 1, and each parameter of the simulation operation increases by 0.1. The specific configuration of scenario 1 is shown in Table 2. Figure 4 shows the network nodes’ deployment obtained when the coverage connectivity factor *ρ* takes different values.

The results with different coverage connectivity factor *ρ* of simulation scenario 1 shown in Table 2 are shown in Figure 5, Figure 6, Figure 7 and Figure 8.

As shown in Figure 5, when the *ρ* of DCGA increases from 0, more neighbors will be admitted into DBP, and the number of static nodes in the network will increase gradually. This is because the connectivity gains between the node and its neighbors gradually increase with the increase of *ρ* in the node’s utility function. Again, as the number of nodes increases, network connectivity has also been significantly improved. Especially as *ρ* is greater than 0.55, the increase of the number of nodes brings about more obvious changes in connectivity probability. In summary, the network connectivity probability can be significantly improved by adjusting the *ρ* of the DCGA, i.e., when the initial node set is large enough, the network node density can be changed by adjusting *ρ*. And proper value of *ρ* can also balance the relationship between network connectivity and economy.

As shown in Figure 6, when the *ρ* of the DCGA increases from 0, the message delay of decreases gradually as more neighbors are admitted into the DBP. Due to the improvement of network connectivity, the message delivery ratio increases with the increase of *ρ*. In addition, with the increase of the number of nodes and the improvement of network connectivity, the network gradually transits from delay-tolerant network to real-time network. Especially when the coverage connectivity factor is greater than 0.6, the increase of *ρ* brings more obvious changes in the message delay. To sum up, the message delay can be significantly improved by adjusting the CCF of DCGA. That is, when the initial set of nodes is large enough, the network node density is changed so that the nodes are reasonably arranged as needed. As *ρ* is large enough, the ability of network to transmit messages tends to be real-time.

As shown in Figure 7, when the coverage connectivity factor of DCGA increases from 0, the network hop count increase with the increase of node density, and the network overhead increases with more neighbors being admitted into DBP. In addition, with the increase of the number of nodes and the improvement of network connectivity, nodes in the network try to improve the delivery ratio of messages by more forwarding, resulting in a very high number of network hop count. Especially when the *ρ* is greater than 0.6, the increase of *ρ* leads to the steady change of network overhead. To sum up, by adjusting the coverage connectivity factor of DCGA, the real-time performance of the network is obviously improved, but at the same time, the network overhead and hop count performance are poor.

As shown in Figure 8, when the *ρ* of DCGA increases from 0, the network lifetime decreases with the increase of node density and message delivery ratio, and the average energy consumption of each node changes little with the change of *ρ*. In addition, with the increase of the number of nodes and the improvement of network connectivity, the nodes in the network try to increase the delivery ratio of messages through more message forwarding, and the sharp increase of network hop count will inevitably lead to the decrease of network lifetime. Especially when the coverage connectivity factor is greater than 0.6, the increase of *ρ* makes the network lifetime decrease significantly. In summary, by adjusting the coverage connectivity factor of the Distributed Coverage Game Algorithm (DCGA), the real-time performance of the network is obviously improved, while the energy consumption of individual nodes is not significantly changed, but the network lifetime is greatly affected.

### 5.2. Performance of DCGA with Different Parameters of α

In order to determine the impact of network energy consumption factor α on network performance, this paper designs simulation scenario 2 to adjust the value of α, and to find the appropriate optimal configuration. In simulation scenario 2, the default value of α is changed. The *α* parameter values vary from 0 to 1, and each simulation run increases by 0.1. The specific configuration of scenario 2 is shown in Table 3. Figure 9 shows the deployment of network nodes obtained when network energy consumption factor α takes different values.

The results with network energy consumption factor *α* of simulation scenario 2 shown in Table 3 are shown in Figure 10, Figure 11, Figure 12 and Figure 13.

As shown in Figure 10, when the energy consumption factor *α* of DCGA increases from 0, more neighbors are admitted into DBP, and the number of static nodes in the network increases gradually. Again, as the number of nodes increases, the connectivity of the network has also been significantly improved. Especially, the alpha of the network is between 0.2 and 0.55, the increase of the number of nodes brings more obvious changes in connectivity probability. When the energy consumption factor is too high or too low, the change of network connectivity probability and number of nodes is relatively mild.

The above phenomena are mainly due to the fact that the ratio of the connectivity weight of the utility function is changing, but the connectivity factor *ρ* is constant. This makes the regulation ability of network density limited.

In conclusion, the connectivity probability can be changed by adjusting the *α* of DCGA, that is, the density of network nodes can be changed when the initial nodes set is large enough, but the adjusting range is limited.

As shown in Figure 11, when *α* of DCGA increases from 0, the message delivery ratio of nodes with improved network connectivity increases with the gradual increase of *α*, more neighbors are accepted into DBP, and the number of static nodes in the network increases gradually. With the improvement of network connectivity, the message delivery ratio of nodes increases gradually. As more neighbors are included in DBP, the number of static nodes increases gradually in the network. In this way, as the number of nodes increases, the network gradually changes from delay tolerant network to real-time network.

Especially, *α* is between 0.2 and 0.7, and the increase of the number of nodes brings more obvious changes in network delay. However, when the factor is too high or too low, the change of message delivery ratio and message delay is gentler. This is mainly because the connectivity factor of the network is constant, which limits the energy-saving ability of the network. In summary, the network connectivity probability can be adjusted by changing of *α*, i.e., *α* can change the real-time performance of the network when the initial node number set is enough.

As shown in Figure 12, when *α* of DCGA increases from 0 to 1, the network overhead increases gradually with the increase of the node density and the hop count in the network.

In addition, with the improvement of network connectivity, nodes in the network attempt to increase the delivery ratio of messages through more message forwarding, resulting in a sharp increase in the number of network hop count. Especially when *α* is greater than 0.7, the increase of value brings the change of network overhead to a steady state. To sum up, by adjusting the *α* of the DCGA, the real-time performance of the network is improved obviously, and the performance of network overhead and network hop count is worse.

As shown in Figure 13, when the *α* of DCGA increases from 0, the nodes themselves tend to adopt the strategy of losing energy performance, and the network lifetime decreases with the increase of *α*. With the change of *α*, the average energy consumption of each node in the network also gradually increases. In addition, with the increase of the number of nodes and the improvement of network connectivity, nodes try to increase the delivery ratio of messages by more message forwarding, and the sharp increase of network hop count will inevitably lead to the decline of network lifetime. Especially after the *α* is greater than 0.2, the lifetime of the network decreases significantly with the increase of network energy consumption factor. To sum up, the network lifetime can be controlled by adjusting the energy consumption factor alpha to change the energy consumption performance of network nodes.

### 5.3. Performance of DCGA Compared with other Algorithms

In order to investigate the influence of different maximum number of nodes on several barrier coverage algorithms, Scenario 3 is constructed. The scenario has changed one or two parameters from the default values, the maximum number of static nodes in the network increases from 5 to 200. Then, the *ρ* of DCGA are set to 0.5 and 1, respectively, to verify the different performance of DCGA in delay tolerant network and real-time network, the *α* of DCGA are set to 0.7 to balance network lifetime and other performances. The barrier coverage algorithms used for comparison in Scenario 3 are TMFA [29] and DP [42]. The 95% confidence intervals were calculated for every result and they were plotted on the figures. The parameters in Scenario 3 are configured, as shown in Table 4, 200 simulations were performed for this scenario. The results for Scenario 3 are shown in Figure 14 and Figure 15.

Results of Scenario 3 show that the proposed method performs as well as the other barrier coverage algorithms in most cases or even better in some respects such as the delivery ratio and delivery time (DCGA, *ρ* = 1.0) or the energy consumption (DCGA, *ρ* = 0.5). 

We can see from Figure 14a,b, in Scenario 3, that the delivery ratio and delivery time act as an increasing function and decreasing function of all parameters for all algorithms, respectively. The message delivered in the proposed (DCGA, *ρ* = 1.0) increases as the parameters increase, followed by TMFA and DP. Similarly, the energy consumption acts as an increasing function of all parameters for all algorithms in Scenario 3. DCGA (*ρ* = 0.5) shows the minor energy consumption as the parameters increase, followed by DCGA (*ρ* = 1.0) and TMFA (Figure 15a). Then in Figure 15b, DCGA (*ρ* = 0.5) shows the maximum number of pending DBPs, this means that it has the strongest network coverage and the best fault tolerance among all algorithms.

### 5.4. Analysis of Simulation Results

Simulation results are analyzed and discussed in this section. The analysis is divided into two parts, i.e., analysis of the change results of DCGA with the different parameters and analysis of the change results of algorithms with the different parameters.

#### 5.4.1. Analysis of the Results of DCGA with Different Parameters

For DCGA, adjusting *ρ* means changing the real-time requirements of the network. In sensor network applications with different message delivery time requirements, such as delay-tolerant network (DTN) and real-time network (RTN) applications, network economy and data timeliness have different requirements. As shown in Figure 5, Figure 6, Figure 7 and Figure 8, with the increase of *ρ*, networks can gradually move from DTNs to RTNs, and the economy of networks is reduced, but the timeliness of networks is significantly improved. It proves that this method can adapt to all kinds of networks with different data delivery time requirements, improve the controllability of network delay in delay-tolerant sensor networks, and balance the relationship between the economy and efficiency of network. In view of the application of robot dynamic monitoring network for smart gird in this paper, delay data can be used for transmission line fault diagnosis, but low data transmission delay and delay controllable network are conducive to improving the level of power grid monitoring. Based on the economy and efficiency of balanced network construction, the value of *ρ* can be set to 0.5–0.8, as show in Figure 5, Figure 6, Figure 7 and Figure 8.

In the practical application of WSN, network energy consumption and lifetime are important indicators. The DCGA proposed in this paper can change the direct relationship between network energy consumption and other performances by adjusting *α*, as shown in Figure 10, Figure 11, Figure 12 and Figure 13. Considering the economy and practicability of network construction, setting *α* to 0.6–0.8 can effectively balance network performance and network lifetime.

#### 5.4.2. Analysis of the Results of Algorithms with the Different Parameters

Adjusting the maximum number of nodes means changing the results of optimal coverage of the network. For all coverage algorithms, message delivery ratio and delivery time of the network are positively correlated with the maximum number of nodes. When the maximum number of nodes can satisfy the real-time network configuration, because DCGA controls the network coverage connectivity through *ρ*, *ρ* = 0.5 and *ρ* = 1.0 show a great difference in network deployment. The former aims at building DTN, and the algorithm tends to select a small number of nodes to build the network; the latter aims at building RTN, and the network performance is better, but the economy is lower. However, TMFA and DP can only construct real-time networks. When the maximum number of nodes is more than 100, the performances of the algorithms are like that of DCGA (a = 1.0).

## 6. Conclusions

This paper proposes a new method to improve network coverage and improve the performance of delay-tolerant networks (DTNs). The main conclusions are as follows:(1)Network dynamic barrier coverage (DBC), as a new method to study linear dynamic delay tolerance, achieves network performance optimization. By adjusting the deployment density of static nodes and reasonably controlling the number of nodes according to the dynamic barrier path (DBP), the network delay and connectivity are greatly improved. By extending the traditional barrier coverage theory and applying the wireless sensor network to transmission line, the dynamic barrier coverage problem can well describe the dynamic sensing coverage characteristics and requirements of linear areas such as transmission lines, rivers and highways.(2)The proposed method mainly includes two parts: dynamic barrier coverage (DBC) model and distributed coverage game algorithm (DCGA). The first part introduces the basic concepts of dynamic barrier coverage problem, and introduces the analysis method of network coverage graph and theoretical model dynamic barrier path. Then, based on the principle of economical deployment of network nodes in transmission line monitoring system, the method of adjusting network connectivity and real-time by calculating connectivity probability in dynamic barrier path is proposed. In the second part, a new network coverage method based on potential game is proposed. The game method fully considers the characteristics of transmission lines and robots, and designs a utility function based on network connectivity probability and network energy consumption. Moreover, a new network coverage algorithm DCGA is designed. This algorithm uses a distributed learning method to calculate and update the utilities of each node, and then the node performs deployment actions according to the utilities combined with the requirements of the application scenario.(3)In the simulation, three scenarios are implemented, and eight related network performance parameters are collected and analyzed. The results show that DCGA can adjust many network performances of dynamic monitoring sensor network for transmission line. Then, the financial cost of network construction is also considered. DCGA can adjust the network connectivity (node deployment density) and network delay by preset coverage connectivity factors according to the requirements of application scenarios, thus balancing the network performance and the cost of the monitoring system.

Further research should focus on the following aspects. Firstly, the relevant parameters of DCGA need to be estimated in advance, and the pre-estimation does not establish a clear basis for setting. It needs to judge the correctness of the preset parameters by setting up a comparative test method. It will take a lot of time and energy. Secondly, DCGA increases network overhead and hop count while improving network performance and connectivity. It is necessary to further improve the formation process of dynamic barrier path and reduce the over-density of nodes under high connectivity. Finally, this part needs further field verification based on the actual project.

## Figures and Tables

**Figure 1 sensors-19-00041-f001:**
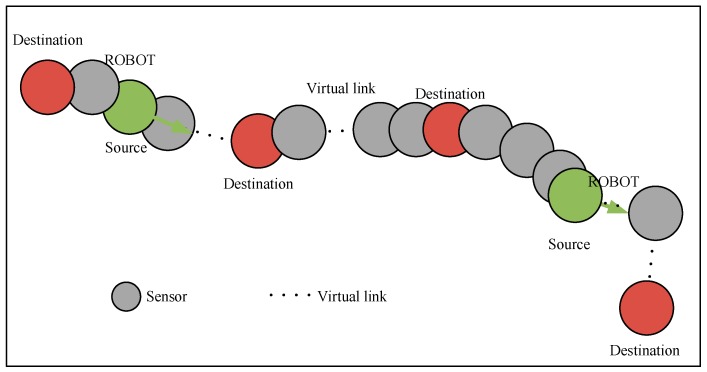
A typical dynamic barrier coverage problem.

**Figure 2 sensors-19-00041-f002:**
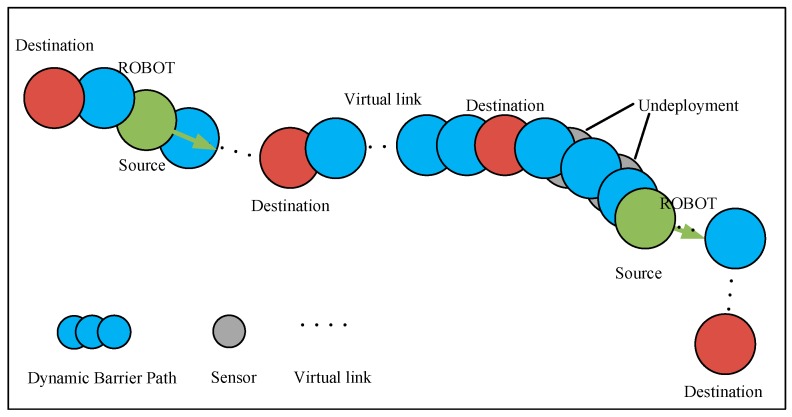
A typical coverage graph of Dynamic Barrier Coverage (DBC).

**Figure 3 sensors-19-00041-f003:**
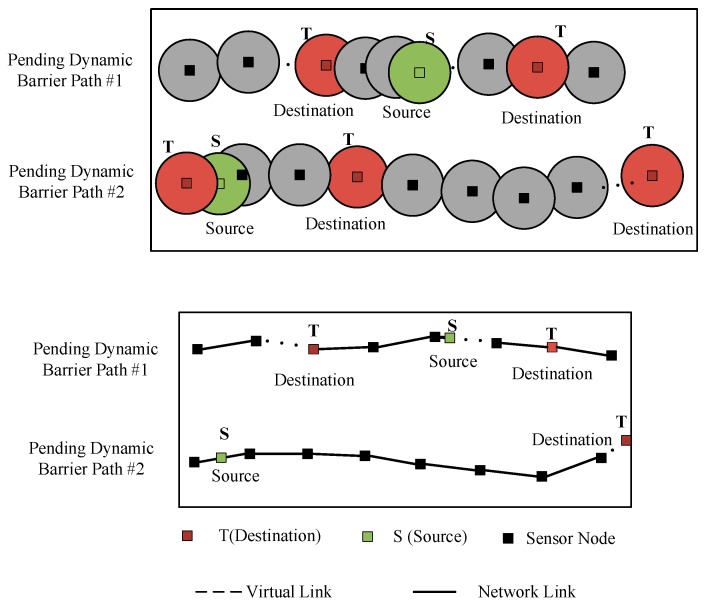
Pending dynamic barrier paths in coverage graph.

**Figure 4 sensors-19-00041-f004:**
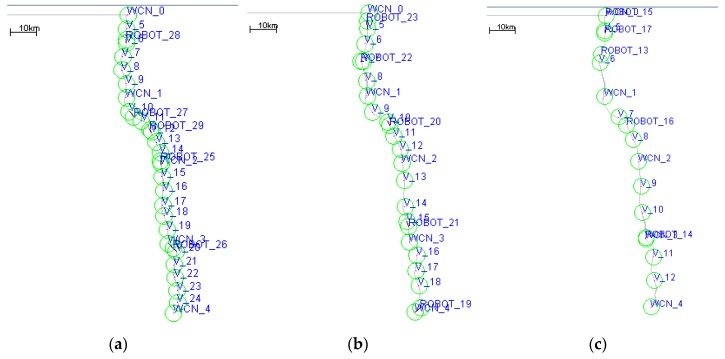
Nodes deployment diagram with different *ρ*. (**a**) *ρ* = 0.9; (**b**) *ρ* = 0.5; (**c**) *ρ* = 0.3.

**Figure 5 sensors-19-00041-f005:**
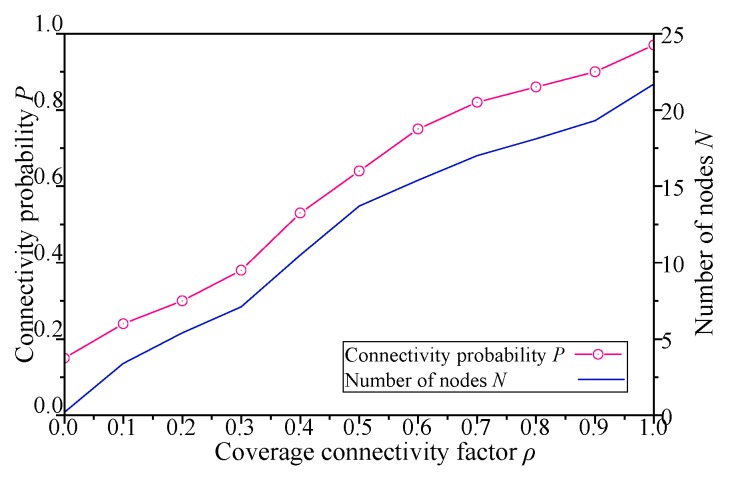
Influence of *ρ* on network connectivity probability and number of nodes.

**Figure 6 sensors-19-00041-f006:**
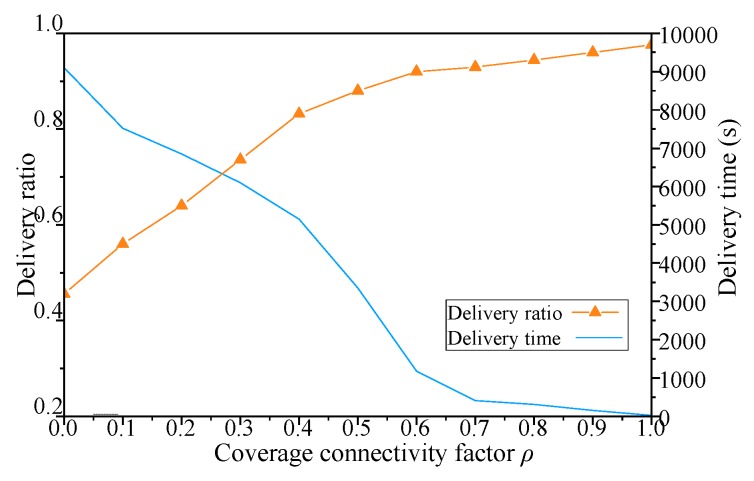
Influence of *ρ* on message delivery ratio and delivery time.

**Figure 7 sensors-19-00041-f007:**
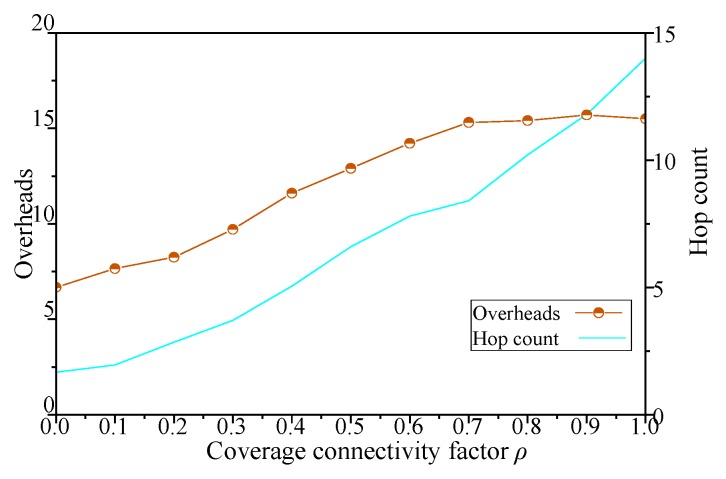
Influence of *ρ* on network overhead and hop count.

**Figure 8 sensors-19-00041-f008:**
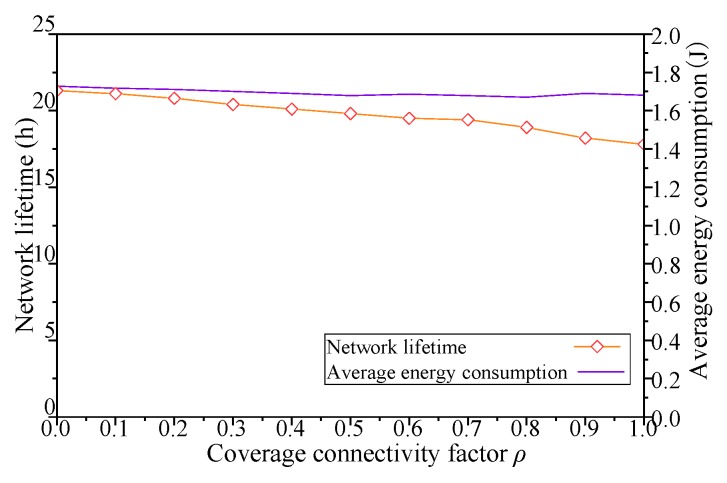
Influence of *ρ* on network lifetime and average energy consumption.

**Figure 9 sensors-19-00041-f009:**
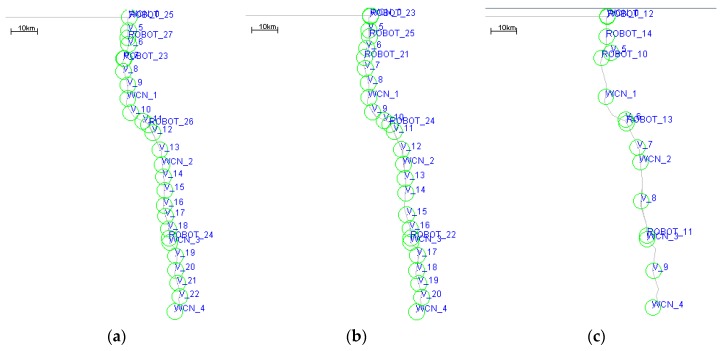
Nodes deployment diagram with different *α*. (**a**) *α* = 0.9; (**b**) *α* = 0.5; (**c**) *α* = 0.3.

**Figure 10 sensors-19-00041-f010:**
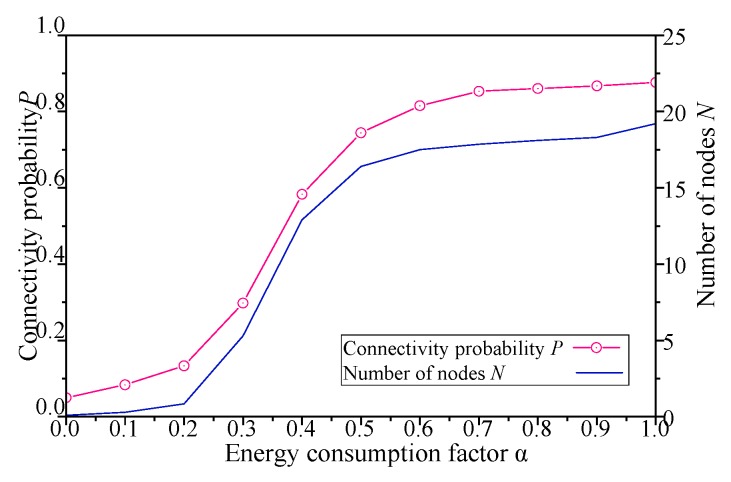
Influence of *α* on network connectivity probability and number of nodes.

**Figure 11 sensors-19-00041-f011:**
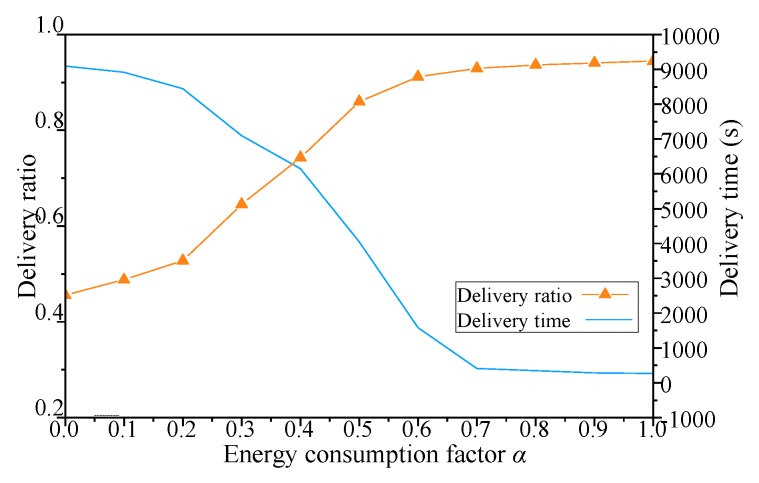
Influence of *α* on message delivery ratio and delivery time.

**Figure 12 sensors-19-00041-f012:**
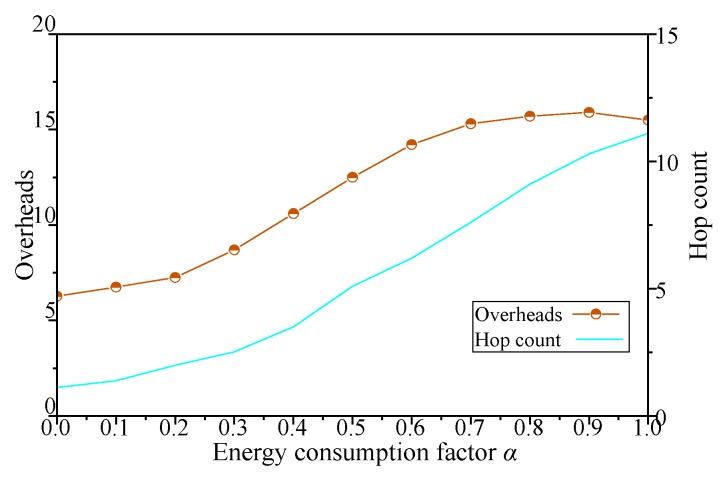
Influence of *α* on network overhead and hop count.

**Figure 13 sensors-19-00041-f013:**
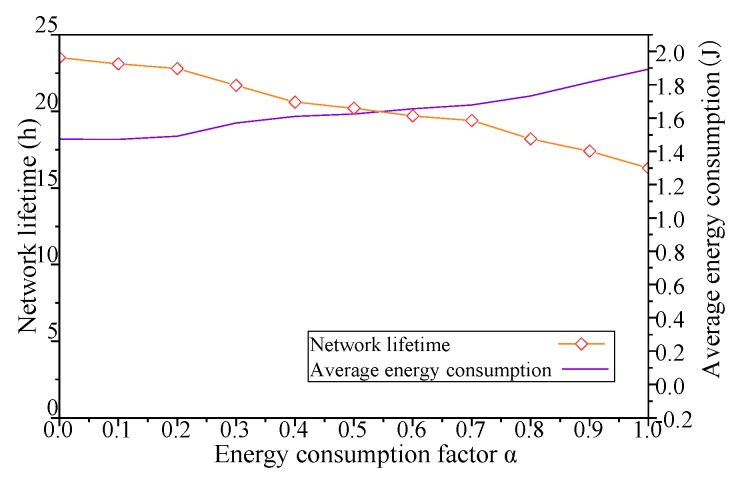
Influence of *α* on network lifetime and average energy consumption.

**Figure 14 sensors-19-00041-f014:**
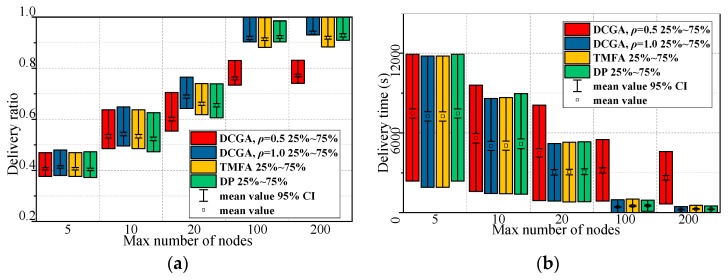
Results of Scenario 3. (**a**) Delivery ratio with different maximum number of nodes; (**b**) delivery time in seconds with different maximum number of nodes.

**Figure 15 sensors-19-00041-f015:**
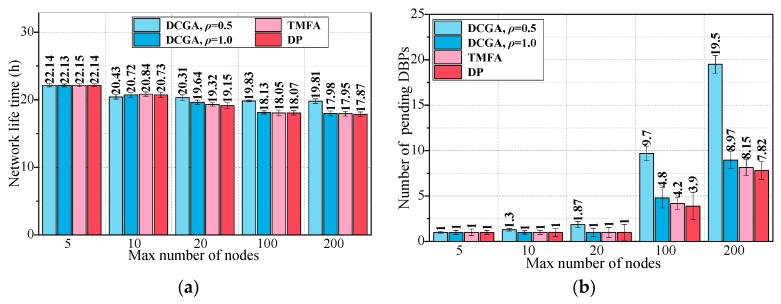
Results of Scenario 3. (**a**) Network life time with different maximum number of nodes; (**b**) number of pending DBPs with different maximum number of nodes.

**Table 1 sensors-19-00041-t001:** Default parameters of the simulation.

Parameter	Default Value
Network area size (m × m)	74,500 × 143,400
Simulation run time (s)	86,400
Number of Robots	5 nodes
Max number of SNs	100 nodes
Number of WCNs	5 nodes
*Rc* of WCNs, SNs and robot (m)	3000
Movement rate of Robot (m/s)	3–5
*T_pause_* (s)	10–30
Buffer size (M)	500
Message size (Kb)	2000
Robot message interval (s)	60, 120
Source nodes	Robot
Destination nodes	WCNs
Movement model	BPRW
Alpha and Beta of DTMA	0.78, 0.22
*ρ* of DCGA	0.8
*α* of DCGA	0.7

**Table 2 sensors-19-00041-t002:** Parameters of Scenario 1.

Parameter	Parameter Variation Range	Parameter Variation Amplitude
*ρ*	0–1	0.1

**Table 3 sensors-19-00041-t003:** Parameters of Scenario 2.

Parameter	Parameter Variation Range	Parameter Variation Amplitude
*α*	0–1	0.1

**Table 4 sensors-19-00041-t004:** Parameters of Scenario 3.

Parameter	Number 1	Number 2	Number 3	Number 4	Number 5
Maximum number of nodes	5	10	20	100	200

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
