# Peer review of "Dynamic Barrier Coverage in a Wireless Sensor Network for Smart Grids"

_sensors, 2018, doi:10.3390/s19010041_

Round 1

Reviewer 1 Report

This well written paper but it will need to improve.

1) Compare your work with the existing method.

2) What are the physical significants of the proposed method?

3) Simulation setting will need to describe  more.

Reviewer 2 Report

In this paper, the authors try to answer the question: how to integrate inspection robots into wireless sensor networks? The reviewer values this work as scientifically sound and novel. Only minor concerns are as follows. 

1. there is no performance comparison. the authors should compare their work to state-of-the-art work from other authors.

2. The reviewer thinks that Fig 4 is not necessary. 

Round 2

Reviewer 2 Report

I am glad to proceed this manuscript to publication stage. 

Sensors EISSN 1424-8220 Published by MDPI AG, Basel, Switzerland RSS E-Mail Table of Contents Alert
Back to Top